# The Impact of Lung Cancer in Patients with Combined Pulmonary Fibrosis and Emphysema (CPFE)

**DOI:** 10.3390/jcm12031100

**Published:** 2023-01-31

**Authors:** Xiaoyi Feng, Yishan Duan, Xiafei Lv, Qinxue Li, Binmiao Liang, Xuemei Ou

**Affiliations:** 1Department of Respiratory and Critical Care Medicine, West China Hospital of Sichuan University, Chengdu 610041, China; 2Department of Radiology, West China Hospital of Sichuan University, Chengdu 610041, China

**Keywords:** combined pulmonary fibrosis and emphysema syndrome (CPFE), lung cancer (LC), acute exacerbation (AE), idiopathic pulmonary fibrosis (IPF), risk factor

## Abstract

Given the high risk of lung cancer (LC) in patients with combined pulmonary fibrosis and emphysema (CPFE), and the difficulty of early diagnosis, it is important to understand the impact of LC in these patients. The effect of LC on the development of acute exacerbation (AE) as a natural course of CPFE is still unknown. We retrospectively reviewed medical records of patients at the West China Hospital and enrolled 59 patients with CPFE combined with LC and 68 CPFE patients without LC for initial diagnosis matched in the same period. We compared the clinical characteristics and imaging features of CPFE patients with LC and without LC, and analyzed the associated factors for the prevalence of LC using binary logistic regression. Cox proportional hazards regression analysis was performed to explore risk factors of AE as a natural course of CPFE. Patients with CPFE combined with LC were more common among elderly male smokers. The most common pathological type of tumor was adenocarcinoma (24/59, 40.7%) and squamous cell carcinoma (18/59, 30.5%). Compared with those in the without LC group, the proportions of men, and ex- or current smokers, and the levels of smoking pack-years, serum CRP, IL-6, fibrinogen, complement C3 and C4 in patients with LC were significantly higher (*p* < 0.05). There was no significant difference in the proportion of natural-course-related AE (10.2% vs. 16.2%, *p* > 0.05) between the two groups. Logistic regression analysis demonstrated that pack-years ≥ 20 (OR: 3.672, 95% CI: 1.165–11.579), family history of cancer (OR: 8.353, 95% CI: 2.368–10.417), the level of fibrinogen > 4.81 g/L (OR: 3.628, 95% CI: 1.403–9.385) and serum C3 > 1.00 g/L (OR: 5.299, 95% CI: 1.727–16.263) were independently associated with LC in patients with CPFE. Compared to those without AE, CPFE patients with AE had significantly higher levels of PLR and serum CRP, with obviously lower DLCO and VC. The obviously increased PLR (HR: 3.731, 95% CI: 1.288–10.813), and decreased DLCO%pred (HR: 0.919, 95% CI: 0.863–0.979) and VC%pred (HR: 0.577, 95% CI: 0.137–0.918) rather than the presence of LC independently contributed to the development of natural-course-related AE in patients with CPFE. Pack-years, family history of cancer, the levels of fibrinogen and serum C3 were independently associated with LC in patients with CPFE. The presence of LC did not significantly increase the risk of AE as a natural course of CPFE. Clinicians should give high priority to CPFE patients, especially those with more severe fibrosis and systemic inflammation, in order to be alert for the occurrence of AE.

## 1. Introduction

Combined pulmonary fibrosis and emphysema syndrome (CPFE) proposed by Cottin et al. is diagnosed by upper lobe emphysema and lower lobe pulmonary fibrosis, characterized by severe dyspnea and significant diffusing dysfunction, mainly in elderly smoking males [1,2,3]. A high proportion of CPFE patients have shown a usual interstitial pneumonia (UIP) pattern on chest high-resolution computed tomography (HRCT) and the presence of the UIP pattern is an important variable associated with death in patients with CPFE [4,5,6,7,8]. Lung cancer (LC) is one of the most common complications of CPFE [2,3]. Previous studies have indicated a significantly higher prevalence of lung cancer in CPFE patients than in patients with normal lung or chronic obstructive pulmonary disease (COPD) [9]. In addition, it was exhibited that patients with CPFE had more than twice the risk of developing lung cancer than patients with idiopathic pulmonary fibrosis (IPF) alone [10]. Much more importantly, the majority of lung cancer patients with CPFE have been found at an advanced stage of the tumor, with worsening overall survival and higher mortality than IPF or COPD patients with lung cancer alone [11,12]. The ‘triple hit’ of smoking, emphysema and pulmonary fibrosis may have contributed to this phenomenon, but it is more likely to be linked to the difficulty of timely detection and diagnosis of lung cancer on chest CT as tumors may be hidden by the involvement of concomitant parenchymal fibrosis and emphysema [13]. An in-depth understanding of the impact of lung cancer in CPFE patients is helpful to identify high-risk groups early, immediately taking more rigorous follow-up and treatment if necessary. 

Acute exacerbation (AE) is defined as a sudden aggravation of dyspnea with new bilateral pulmonary infiltrates, with no evidence of other known causes of deteriorating respiratory function [14]. AE of CPFE often leads to acute adverse events such as respiratory failure and severe infections, and even requires emergency hospitalization, usually with a poor prognosis [15]. A previous study found that AE occurred in 24% of CPFE patients during follow-up [16]. So far, it has been believed that low forced vital capacity (FVC), low carbon monoxide diffusion capacity (DLCO), low 6-min walking distance (6MWD), severe dyspnea, elevated serum level of Krebs von Lungen-6 (KL-6), increased peripheral monocyte count and so on have been proved to be risk factors for AE of IPF [17,18,19,20]. However, the effect of LC on AEs in patients with CPFE remains to be unclear. Several studies have shown that in patients with CPFE combined with LC, there is a significantly increased risk of AE triggered by oncologic treatments such as chemotherapy and surgery, especially in the Asian population [9,11,21]. Yet so far the effect of LC on the occurrence of AE associated with the natural course of CPFE has not been well elucidated. 

This study systematically compared the differences in clinical characteristics and imaging features between CPFE patients with and without lung cancer. More importantly, we further revealed the predictive risk factors for the development of AE as a natural course of CPFE progression.

## 2. Materials and Methods

### 2.1. Study Population

This was a retrospective and observational study, approved by the Biomedical Ethics Review Committee of the West China Hospital of Sichuan University (No. 2021-1374). Fifty-nine patients with CPFE combined with LC admitted to the Department of Respiratory and Critical Care Medicine from October 2017 to September 2020 and 68 CPFE patients without LC matched in the same period were enrolled. All patients included were newly diagnosed with CPFE. Patients with connective tissue disease, autoimmune disease, pure asthma, severe cardiovascular and cerebrovascular diseases, hematologic tumor, kidney failure and other diseases that may affect the relevant clinical indicators were excluded, as well as patients without confirmed primary lung cancer. 

### 2.2. Definition of Patients with CPFE

Patients were screened according to the diagnostic criteria based on chest HRCT for CPFE proposed by Cottin et al. [15], which are as follows: (1) Upper lobe emphysema of any subtype defined as well-demarcated areas of low attenuation (CT value < 910HU) delimitated by a very thin wall (<1 mm) or no wall, and emphysema area/total lung volume > 5%. (2) Lower lobe showed lung fibrosis of any subtype. 

In this study we excluded patterns of lung fibrosis different from usual interstitial pneumonia (UIP) including typical grid shadow, honeycombing and traction bronchiectasis in subpleural and lower lung distribution. The UIP was diagnosed on the presence of a definite or probable UIP pattern based on the 2018 ATS/ERS/JRS/ALAT official IPF diagnosis and management guidelines [22]. All the diagnoses were ultimately made by two senior respiratory physicians and one radiologist on the basis of comprehensive consideration of clinical symptoms and imaging findings (Figure 1).

### 2.3. Diagnosis of Lung Cancer

All patients with lung cancer were diagnosed by pathological examination or surgery. Staging of lung cancer was performed in accordance with the 2015 National Comprehensive Cancer Network guidelines. 

### 2.4. Definition of AE

All patients were followed up by telephone, outpatient and inpatient visits in November, 2022. The median follow-up period was 42.1 months (range 13.6–72.4 months). We defined AE as a natural course of CPFE by the following criteria [23]: (1) sudden deterioration or development of dyspnea typically within 30 days, (2) new bilateral ground-glass opacity and/or consolidation superimposed on a background pattern consistent with UIP pattern on chest radiograph and (3) with no evidence of other known causes such as cardiac failure or fluid overload of deteriorating respiratory function. (4) Acute deterioration triggered by anti-cancer drugs (carboplatin/etoposide, carboplatin/paclitaxel/bevacizumab, gefitinib) (up to 4 weeks after treatment) and post-operative AE (up to 12 months after surgery) were excluded from CPFE natural-course-related AE. 

### 2.5. Data Collection 

Data on each patient was retrospectively extracted from the electronic medical records, including baseline clinical characteristics, underlying diseases, the initial laboratory examinations when diagnosed newly, pulmonary function test, CT imaging and ultrasonic cardiogram features, pathological type, location and stage of the cancer. Smoking levels were expressed in pack-years calculated by multiplying the number of packs of cigarettes consumed per day by the number of years smoked. Forced expiratory volume in the first second (FEV1%), forced expiratory volume in the first second to forced vital capacity ratio (FEV1/FVC%), vital capacity (VC, %pred) and diffusing capacity for carbon monoxide (DLCO, %pred) were performed in pulmonary function tests. We evaluated the severity of pulmonary fibrosis by using the composite physiological index (CPI) in patients with CPFE. The formula for calculating CPI value was 91—(0.65 × DLCO%)–(0.53 × FVC%) + (0.34 × FEV1%).

## 3. Statistical Analysis

The analysis was performed with SPSS 24.0 (SPSS Inc., Chicago, IL, USA). Qualitative variables were expressed as counts and frequencies, and quantitative variables were expressed as the median and interquartile range (IQR). For qualitative variables, we chose to use Fisher’s exact test or χ^2^ test depending on the data. Quantitative variables were compared by Mann–Whitney U test. Binary logistic regression analysis was performed to identify the association between clinical factors and LC in CPFE patients and odds ratios (OR) and 95% confidence intervals (95% CIs) were estimated. Receiver operating characteristic (ROC) curves were constructed to determine the optimal cut-off values for clinical indicators by the Youden index. The allocation was largely based on established cut-offs. We used ROC curve analysis to assess the sensitivity and specificity of the predictive model for the occurrence of lung cancer in CPFE patients. The concordance index was used to assess the accuracy of the model, which ranged from 0.5 to 1.0—higher values indicated higher discriminatory power. Cox proportional hazards regression analyses were used to evaluate the impact of laboratory parameters and clinical characteristics on AE. Variables with *p* < 0.1 in univariate analysis were included in multivariate analysis. In this study, *p* < 0.05 was considered as statistically significant. 

## 4. Results

### 4.1. Baseline Clinical Characteristics and Tumor Features in CPFE Patients with Lung Cancer

Patients with CPFE combined with lung cancer were more common among elderly male smokers, especially heavy smokers. The most common pathological types of the tumor were adenocarcinoma (24/59, 40.7%) and squamous cell carcinoma (18/59, 30.5%). Tumors associated with CPFE were more inclined to occur in the lower lobe and peripheral of the lung, with a greater tendency to be in the fibrosis areas in marked contrast to emphysema areas. The majority of patients had a low degree of tumor differentiation and were at an advanced stage at the time of first diagnosis (Table 1). 

### 4.2. Comparison between CPFE Patients with LC and without LC 

The proportion of men (100.0% vs. 86.8%, *p* < 0.05), ex- or current smokers (84.7% vs. 67.6%, *p* < 0.05) and the level of smoking pack-years (40 (20, 45) vs. 20 (0, 45), *p* < 0.05) in the lung cancer group were significantly higher than those in the without lung cancer group. Among CPFE patients with LC, the proportion of patients with previous tuberculosis (22.0% vs. 10.3%, *p* < 0.05) and family history of LC (18.6% vs. 8.8%, *p* < 0.05) was significantly higher than in the group without LC. Moreover, the levels of ALC (1.36 (0.99, 2.01) vs. 1.13 (0.80,1.63)), CRP (57.30 (9.80, 90.30) vs. 35.40 (8.51, 80.42)), IL-6 (70.30 (11.57, 101.99) vs. 42.53 (12.93, 64.47)), fibrinogen (4.95 (3.66, 5.66) vs. 3.97 (3.07, 4.98)), complement C3 (0.969 (0.852, 1.100) vs. 0.895 (0.765, 0.975)) and C4 (0.235 (0.191, 0.279) vs. 0.204 (0.163, 0.246)) in LC patients were much higher than those in patients without cancer (*p* < 0.05). There were no statistical differences in the levels of NLR and PLR between the two groups (*p* > 0.05).

On pulmonary function test the mean FEV1/FVC ratios in the lung cancer group were significantly lower than those in the without lung cancer group (77.4 (72.8, 81.4) vs. 82.0 (72.3, 93.7), *p* < 0.05). No statistically significant differences were observed in terms of CPI and DLCO%pred between the two groups. Regardless of the combination of lung cancer, the typical honeycombing sign and traction bronchiectasis and paraseptal emphysema were the most common imaging features of pulmonary fibrosis and emphysema, respectively, on HRCT in CPFE with UIP patients (Table 2). There was no significant difference in the proportion of the natural-course-related AE of CPFE (10.2% vs. 16.2%, *p* > 0.05) during follow-up in the lung cancer group compared to without lung cancer group. 

### 4.3. Independent Factors for Lung Cancer in Patients Combined with CPFE 

We performed univariate and multivariate logistic regression analysis to explore independent factors for lung cancer in CPFE patients, which demonstrated that pack-years ≥ 20 (OR: 3.672, 95% CI: 1.165–11.579), family history of cancer (OR: 8.353, 95% CI: 2.368–10.417), fibrinogen > 4.81 g/L (OR: 3.628, 95% CI: 1.403–9.385) and serum C3 > 1.00 g/L (OR: 5.299, 95% CI: 1.727–16.263) were independently associated with lung cancer in patients with CPFE (Table 3). Ultimately, the logistic model obtained was statistically significant (*p* < 0.01) and the ROC curve is shown in Figure 2, with an area under the curve (AUC) of 0.844 (95% CI: 0.776–0.913). The sensitivity of this model was 94.9% and the specificity was 63.1% (Figure 2).

### 4.4. Comparison between the Low C3 and High C3 Groups of CPFE Patients with Lung Cancer

Binary logistic regression analysis showed that complement C3 was independently associated with the presence of lung cancer in CPFE patients. Therefore, we further analyzed the clinical and tumor characteristics of patients in the high C3 and low C3 group. Using 1.00 as the cut-off point, the patients with CPFE were divided into 84 cases in the low C3 group and 43 cases in the high C3 group. Patients with a high level of C3 had a significantly higher prevalence of lung cancer (67.4% vs. 35.7%, *p* = 0.009). More importantly, for analysis, CPFE patients with lung cancer were further stratified into a low C3 group (*n* = 29) and another group with high C3 (*n* = 30). Comparing the high C3 group to the low C3 group in lung cancer patients, we found that the levels of serum CRP (82.37 (61.89, 90.91) vs. 51.90 (43.19, 60.21), *p* < 0.05) and IL-6 (89.80 (66.57, 127.13) vs. 53.69 (31.61, 57.68), *p* < 0.05) were significantly higher in the high C3 patients. In addition, the results revealed that patients with high levels of C3 had an increased probability of distant metastases, with brain metastases being more prone, although there was no significant difference (*p* < 0.10) (Table 4). No significant differences were observed in terms of age, gender, smoking status or tumor location, stage or pathological type between the two groups (Table 4).

### 4.5. Comparison between AE and without AE Patients with Lung Cancer

Based on the AE occurring during the period of follow-up (excluding AE caused by chemotherapy or post-surgery), all CPFE patients were divided into a group which developed AE (*n* = 17) and another group without AE (*n* = 110). Underlying disease combinations differed between AE and non-AE patients, with a significantly higher proportion of patients in the AE group having more than two types of underlying disease combined than in the non-AE group (29.4% vs. 9.1%, *p* < 0.05). Compared to those without AE, patients with AE had a significantly higher level of PLR (157.4 (102.2–223.4) vs. 103.4 (75.6–222.9), *p* < 0.05), serum CRP (78.00 (13.20, 121.50) vs. 25.10 (8.51, 80.43), *p* < 0.05), D-dimer (2.33 (0.85–6.31) vs. 1.16 (0.48–2.59), *p* < 0.05) and CD4/CD8 (1.85 (1.10–3.05) vs. 1.33 (0.79–1.89), *p* < 0.05), with obviously lower DLCO% (39.2 (29.0, 46.0) vs. 53.9 (45.8, 69.8), *p* < 0.05) and VC% (75.2 (52.5, 85.9) vs. 85.8 (76.9, 100.8), *p* < 0.05). As for the imaging features there was no significant difference in the presentation of pulmonary fibrosis between the two groups, but the results showed that paraseptal emphysema was more common in the AE group, while in the non-AE group it was predominantly of the centrilobular type. Compared with patients without AE, there was no significant difference in the proportion of lung cancer in CPFE patients with AE (35.3% vs. 48.2%, *p* > 0.05) (Table 5).

### 4.6. Risk Factors for Natural-Course-Related AE in Patients Combined with CPFE

We investigated independent risk factors for the development of natural-course-related AE in CPFE patients by Cox proportional hazards regression. The results demonstrated that increased PLR (HR: 3.731, 95% CI: 1.288–10.813), and decreased DLCO%pred (HR: 0.919, 95% CI: 0.863–0.979) and VC%pred (HR: 0.577, 95% CI: 0.137–0.918) rather than the presence of lung cancer independently contributed to the development of natural-course-related AE in patients with CPFE (Table 6).

## 5. Discussion

This retrospective study resulted in two important findings. First, pack-years, family history of cancer, fibrinogen and serum C3 were independently associated with the prevalence of lung cancer in patients with CPFE. Second, elevated PLR, and decreased DLCO and VC rather than the presence of lung cancer significantly increase the risk of natural-course-related AE in CPFE patients.

Chronic inflammation and lung injury have a remarkable effect on the initiation and progression of tumors, which can be caused by the ‘triple hit’ of smoking, emphysema and fibrosis, resulting in a significantly higher risk of the occurrence of lung cancer in CPFE patients than in patients with IPF or COPD alone [9,10]. Under smoking exposure, airway epithelial cells release pro-inflammatory cytokines to synthesize IL6 and CRP, which activate a series of exaggerated innate immune responses by enhancing immune cell phagocytosis, resulting in further damage to alveoli and promoting tumors through the epithelial–mesenchymal transition (EMT) process [24]. Our research showed that CPFE with lung cancer was more common in elderly men and smokers, mainly in heavy smokers, with a more severe systemic inflammatory response represented by significantly elevated CRP, IL-6 and fibrinogen, which is in line with the results of previous studies [20,21]. It was shown that squamous metaplasia was observed more frequently in and around the foveal region in patients with CPFE lung cancer [25]. Consistent with other studies, in our study the majority of lung cancers in CPFE were prone to occur in the lower lobes and adjacent fibrotic areas, similar to IPF-related lung cancer. In the present study, the histological type of CPFE patients was mainly adenocarcinoma followed by squamous carcinoma, which was inconsistent with some previous studies [12,13,16,17]. The reasons for this phenomenon may be mainly due to racial differences and the increasing proportion of adenocarcinoma in lung cancer in recent years.

The formation and progression of tumors is closely related to the immune status of the host [26]. For a long time, as an important part of innate immunity, the complement system has played a crucial role in immune surveillance of malignant tumors and inhibition of tumorigenesis [27]. However, recent studies have shown that complement can also promote the growth of cancer [28]. Atsuhiko Toyama et al. found that upregulated complement C3 and its fragmentation could be used as biomarkers for lung cancer screening [29]. In this study we discovered that a high level of serum C3 was statistically and independently associated with lung cancer in CPFE patients. The presence of lung cancer in patients with high complement C3 was significantly higher than that in patients with low C3. There are many possible underlying mechanisms linking the complement system and lung cancer. At first, the cleavage product of C3 followed by a series of reactions finally cleaves C5 into C5a which can promote tumor development by recruiting myeloid-derived suppressor cells (MDSCs) [30]. Then, activation products such as C3a and C5a can stimulate the formation of more neovascularization in tumor tissues by increasing the chemotaxis of vascular endothelial cells and inducing local tumor immunosuppression [31]. What is more, Adrienne Boire’s study indicated upregulation of complement C3 in a model of meningeal metastasis of lung and breast cancer and found that cancer-derived C3 activated C3a receptors on choroid plexus epithelial cells to disrupt the blood–brain barrier, proving that C3 was necessary for cancer to grow in the meninges [32]. Interestingly, in agreement with their results, we also found that, compared with patients with low C3, patients with high C3 had a higher probability of distant metastasis, in which the development of brain metastasis seemed to be more likely, although there was no statistically significant difference. Finally, upregulated C3a and C5a have been widely interpreted as indicators of the primary inflammatory response [29,33]. In the present study, we similarly found that CPFE patients with high levels of C3 were usually accompanied by a significantly elevated inflammatory response represented by CRP and IL-6.

The natural history of CPFE is highly heterogeneous, from chronic stable symptoms to progressive respiratory failure or AE, and the incidence varies by race and genetic regulatory factors [15]. Several studies have shown that patients with CPFE combined with lung cancer are at significantly increased risk of AE, a large part of which is due to the significantly higher risk of AE triggered by oncologic treatments such as chemotherapy and surgery, especially in the Asian population [21,34,35]. Otsuka et al. reported that post-surgery-related AE occurred in 13.0% of surgically resected lung cancer patients with CPFE [36]. In Moon’s research it was reported that AE within 1 month after treatment (chemotherapy, surgery or radiotherapy) was more frequent in patients with CPFE than those with IPF alone in lung cancer patients [21]. They found that CPFE (OR: 2.26, 95% CI: 1.09–4.69, *p* = 0.029) showed a significant correlation with AE in patients with NSCLC. Although previous studies have compared CPFE-LC to IPF-LC, few studies have compared CPFE-LC to CPFE. In the study by Jee Youn Oh et al., the proportion of AE in patients with LC combined with CPFE was significantly higher than that in CPFE patients without LC (16.4% vs. 12.7%, *p* < 0.01), with up to 22.0% of them dying from AE among LC patients [11]. They concluded that lung cancer [HR: 3.27, 95% CI 1.44–7.43, *p* < 0.01] was found to be a significant predictor of AE after adjusting for significant variables in patients with CPFE. However, in all of their researches, they mainly studied the occurrence and prognosis of AE related to oncology treatments. The relationship between lung cancer and AE due to the natural course of CPFE during follow-up is still unclear. Excluding AE from chemotherapy and surgical factors in our study, we found that the obviously increased PLR, and decreased DLCO and VC% rather than the presence of lung cancer independently contributed to the development of natural-course-related AE in patients with CPFE. This suggested that the presence of lung cancer did not significantly increase the risk of AE in the natural course of CPFE.

The development of COPD or IPF is closely related to the chronic inflammatory response of the body. It has been well documented that biomarkers such as neutrophil to lymphocyte ratio (NLR), monocyte to lymphocyte ratio (MLR) and platelet to lymphocyte ratio (PLR) correlate more strongly with chronic systemic inflammation than individual cell populations [37,38,39]. More importantly, these biomarkers can be recognized as an immune response due to various stress stimuli and have been shown to be strongly associated with poor prognosis in COPD, IPF, acute myocardial infarction, acute pulmonary embolism and so on [38,39,40,41,42]. Some studies have confirmed that PLR levels are significantly higher in AEs of COPD compared with stable period and decrease gradually after treatment, suggesting that PLR may be related to the severity of COPD. In addition, Mohammad et al. found that the level of PLR was significantly higher in IPF patients with AE compared to those without AE [37]. In the present study we found that increased PLR was a strong independent risk factor for the development of AE in CPFE patients, which requires clinicians to attach great importance to it.

AEs of IPF are more common in patients with physiologically and functionally advanced disease. Pulmonary function measurements are an important method for the diagnosis of IPF and for assessing disease status. In the present study, worse pulmonary function indicators characterized by reduced VC and DLCO were independent risk factors for the development of AE in CPFE patients, which was consistent with the findings of Yasuhiro Kondoh et al. [17,18,43,44]. It has been reported that patients with reduced FVC and VC usually have reduced normal lung area due to extensive fibrosis, and these patients are prone to develop severe lung injury [18,45]. It is recommended that pulmonary function in IPF patients should be checked every 3–6 months.

Therefore, regardless of the combination of lung cancer or not, clinicians should give high priority to CPFE patients, especially those with more severe fibrosis and systemic inflammation, in order to prevent the development of AE.

This study also has a few limitations. First, this was a retrospective, small-sample and single-center study. In addition, this study has not yet classified the severity of emphysema and pulmonary fibrosis, so that it cannot yet demonstrate the correlation between the tumors and the severity of emphysema or fibrosis in CPFE patients. Finally, we did not further attribute the AE occurring in CPFE patients to the emphysema or fibrosis component of the disease such as AE-COPD or AE-IPF. Nevertheless, to the best of our knowledge, this was the first original study to explore the risk factors of lung cancer in CPFE patients, and the relationship between lung cancer and natural-course-related AE in a Chinese population as well. Further prospective studies are needed to explore the mechanism of lung cancer and determine a specific cancer screening program for these patients.

## 6. Conclusions

Pack-years, family history of cancer, and levels of fibrinogen and serum C3 were independently associated with lung cancer in patients with CPFE. The presence of lung cancer did not significantly increase the risk of AE as a natural course of CPFE. Irrespective of the combination with lung cancer or not, clinicians should give high priority to CPFE patients, especially those with more severe fibrosis and systemic inflammation, in order to be alert for the occurrence of AE.

## Figures and Tables

**Figure 1 jcm-12-01100-f001:**
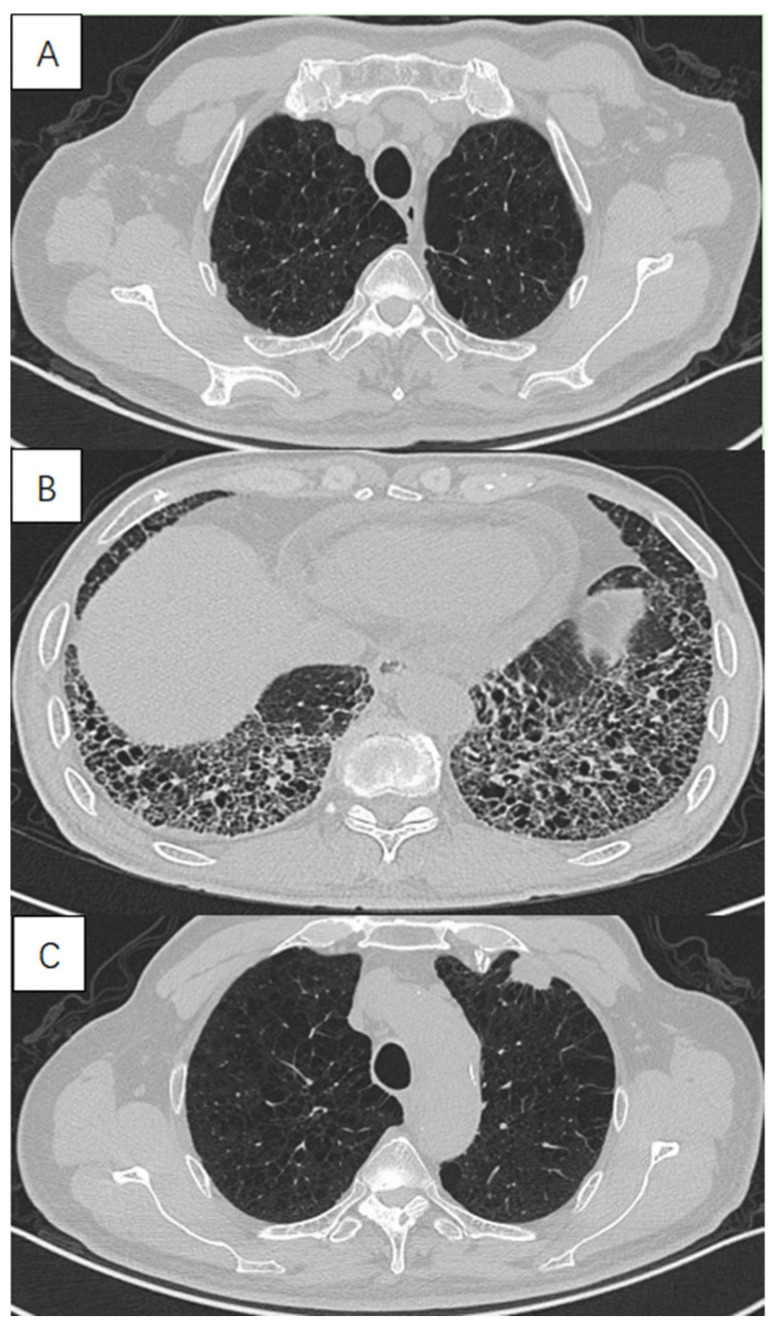
High-resolution computed tomography in a 72-year-old male with lung cancer with combined pulmonary fibrosis and emphysema (CPFE), showing (**A**) paraseptal emphysema in the upper lobe, (**B**) lower-zone-predominant fibrosis and (**C**) a solid mass in the subpleural emphysema area in the left upper lobe.

**Figure 2 jcm-12-01100-f002:**
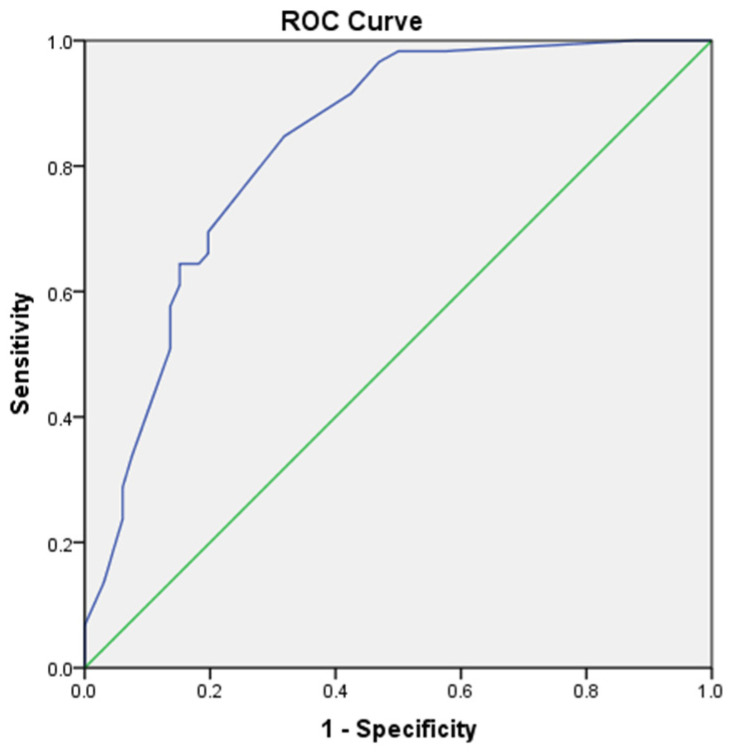
The ROC curve of the logistic model including 4 variables of pack-years ≥ 20, family history of cancer, fibrinogen and C3 is shown, with an AUC of 0.844 (95% CI: 0.776–0.913). The sensitivity of this model was 94.9% and the specificity was 63.1%.

**Table 1 jcm-12-01100-t001:** Baseline clinical characteristics and tumor features of lung cancer patients with CPFE.

	CPFE-LC Group	*n*
Sex (men)	59/59 (100.0%)	59
Age, years	66 (62, 71)	59
Ex- or current smokers	50/59 (84.7%)	59
Pack-years	40 (20, 45)	59
Localization		59
Upper lobe	25/59 (42.4%)	
Lower lobe	34/59 (57.6%)	
Cancer in emphysema areas	26/59 (44.1%)	59
Cancer in fibrosis areas	33/59 (55.9%)	
Central lung cancer	13/59 (22.0%)	59
Peripheral lung cancer	46/59 (78.0%)	
Pathological type		59
Adenocarcinoma	24/59 (40.7%)	
Squamous carcinoma	18/59 (30.5%)	
NOS	7/59 (11.9%)	
Small cell carcinoma	6/59 (10.2%)	
Large cell carcinoma	4/59 (6.8%)	
T 1/2/3/4	7/14/15/23	59
N 0/1/2/3	5/5/19/30	59
M 0/1	29/30	59
Staging I/II/III/IV	3/7/18/31	59
Degree of differentiation		50
Poorly differentiated	27/50 (54.0%)	
Moderately differentiated	15/50 (30.0%)	
Highly differentiated	8/50 (16.0%)	

CPFE-LC: patients with lung cancer (LC) and combined pulmonary fibrosis and emphysema (CPFE); NOS: not otherwise specified; T: tumor; N: node; M: metastasis.

**Table 2 jcm-12-01100-t002:** Clinical characteristics and imaging features in CPFE patients with and without lung cancer.

	CPFE-LC Group	CPFE Group	*p* Value
	*n* = 59	*n* = 68	
Patient characteristics, number (%) or median (Q1,Q3)			
Sex (men)	59/59 (100.0%)	59/68 (86.8%)	0.003
Age, years	66 (62, 71)	71 (65, 76)	0.156
BMI, kg/m^2^	22.66 (20.40, 24.81)	21.30 (19.93, 23.90)	0.149
Ex- or current smokers	50/59 (84.7%)	46/68 (67.6%)	0.038
Pack-years	40 (20, 45)	20 (0, 45)	0.015
Dust exposure	1/59 (1.7%)	0/68 (0.0%)	0.465
Coronary heart disease	10/59 (16.9%)	14/68 (20.6%)	0.189
Diabetes mellitus	13/59 (22.0%)	16/68 (23.5%)	0.841
Previous pulmonary tuberculosis	13/59 (22.0%)	7/68 (10.3%)	0.039
Family history of cancer	11/59 (18.6%)	6/68 (8.8%)	0.042
Laboratory examinations, median (Q1,Q3)			
ANC, ×10^9^/L	5.67 (4.23, 8.61)	5.94 (3.77, 9.54)	0.755
WBC, ×10^9^/L	7.99 (6.78, 11.11)	8.27 (5.77, 11.17)	0.182
ALC, ×10^9^/L	1.36 (0.99, 2.01)	1.13 (0.80, 1.63)	0.018
PLT, ×10^9^/L	197.0 (121.0,248.0)	194.5 (133.5,245.5)	0.667
NLR	3.88 (2.85,6.87)	5.17 (2.59,10.18)	0.173
PLR	128.76 (89.23,198.43)	146.5 (88.08,223.46)	0.124
CRP, mg/L	57.30 (9.80, 90.30)	35.40 (8.51, 80.42)	0.046
IL-6, ug/L	70.30 (11.57, 101.99)	42.53 (12.93, 64.47)	0.043
PCT, ng/mL	0.06 (0.03, 0.15)	0.07 (0.04, 0.16)	0.322
Bilirubin, umol/L	10.40 (7.75, 14.60)	10.85 (6.75, 14.87)	0.985
Cystatin C, mg/L	1.11 (0.93, 1.25)	1.11 (0.91, 1.30)	0.924
Fibrinogen, g/L	4.95 (3.66, 5.66)	3.97 (3.07, 4.98)	0.009
D-dimer, mg/L FEU	1.17 (0.49, 2.86)	1.19 (0.60, 2.79)	0.978
BNP, ng/L	337 (129, 958)	563 (129, 1350)	0.139
C3, g/L	0.969 (0.852, 1.100)	0.895 (0.765, 0.975)	0.017
C4, g/L	0.235 (0.191, 0.279)	0.204 (0.163, 0.246)	0.022
CD4^+^ T cells, cell/µL	438.0 (238.5, 570.5)	414.5 (164.5, 574.7)	0.803
CD8^+^ T cells, cell/µL	243.0 (215.0, 325.0)	312.5 (190.0, 479.7)	0.293
CD4/CD8	1.49 (0.99, 2.16)	1.31 (0.77, 2.00)	0.168
Pulmonary function test, median (Q1,Q3)			
CPI	39.6 (27.1, 50.2)	41.7 (26.4, 55.8)	0.427
FEV1, %pred	86.3 (77.1, 93.5)	79.0 (61.5, 87.5)	0.067
FEV1/FVC, %	77.4 (72.8, 81.4)	82.0 (72.3, 93.7)	0.047
VC, %pred	86.2 (76.9, 99.0)	79.2 (61.8, 87.4)	0.089
DLCO, %pred	42.5 (32.0, 56.1)	45.9 (38.5, 57.8)	0.209
Radiological examinations and occurrence of AE, number (%) or median (Q1,Q3)			
Morphology of pulmonary fibrosis			
Typical honeycombing	29/59 (49.2%)	37/68 (54.4%)	0.258
Reticular pattern	22/59 (37.3%)	21/68 (30.9%)	0.693
Reticular pattern with mild GGO	8/59 (13.5%)	10/68 (14.7%)	0.506
Peripheral traction bronchiectasis or bronchiolectasis	42/59 (71.2%)	43/68 (63.2%)	0.168
Emphysema phenotype			0.256
Paraseptal emphysema	32/59 (54.2%)	37/68 (54.4%)	
Panlobular emphysema	4/59 (6.8%)	2/68 (2.9%)	
Centrilobular emphysema	23/59 (39.0%)	29/68 (42.6%)	
PA/A > 1	26/59 (44.1%)	38/68 (55.9%)	0.211
MPA, mm	23.0 (22.0, 24.0)	26.5 (21.0, 29.0)	0.071
AE	6/59 (10.2%)	11/68 (16.2%)	0.139

CPFE-LC: patients with lung cancer(LC) and combined pulmonary fibrosis and emphysema (CPFE); BMI: body mass index; ANC: absolute neutrophil count; WBC: white blood count; ALC: absolute lymphocyte count; PLT: platelet count; NLR: neutrophil to lymphocyte ratio; PLR: platelet to lymphocyte ratio; CRP: C-reaction protein; IL-6: interleukin-6; PCT: procalcitonin; BNP: brain natriuretic peptide; C3: complement 3; C4: complement 4; CPI: compound physiological index; FEV1: forced expiratory volume in the first second; FVC: forced vital capacity; FEV1/FVC%: forced expiratory volume in the first second to forced vital capacity ratio; VC: vital capacity; DLCO: diffusing capacity for carbon monoxide; GGO: ground-glass opacity; PA/A: pulmonary artery to aorta ratio; MPA: main pulmonary artery; AE: acute exacerbation; Q: quartile.

**Table 3 jcm-12-01100-t003:** Univariate and multivariate analyses of independent factors for the presence of LC in patients with CPFE.

Variable	Univariate Analyses		Multivariate Analyses	
	OR (95% CI)	*p* Value	OR (95% CI)	*p* Value
Sex (men)	1.182 (0.911–2.798)	0.498		
Ex- or current smokers	1.259 (1.012–2.538)	0.045	1.085 (0.261–4.508)	0.911
Pack-years ≥ 20	3.991 (1.893–8.416)	0.000	3.672 (1.165–11.579)	0.026
Previous pulmonary tuberculosis	3.565 (1.103–11.519)	0.034	4.615 (0.714–11.354)	0.375
Family history of cancer	3.149 (1.128–8.817)	0.028	8.353 (2.368–10.417)	0.002
ALC > 1.87 × 10^9^/L	3.362 (1.388–8.142)	0.047	3.439 (0.080–10.951)	0.157
CRP > 75.3 mg/L	1.004 (0.994–1.019)	0.301		
IL-6 > 123.1 ug/L	1.000 (0.988–1.011)	0.892		
Fibrinogen > 4.81 g/L	2.743 (1.441–8.257)	0.049	3.628 (1.403–9.385)	0.008
C3 > 1.00 g/L	2.283 (1.011–6.617)	0.031	5.299 (1.727–16.263)	0.004
C4 > 0.24 g/L	2.315 (1.122–4.779)	0.023	0.690 (0.244–1.954)	0.485
FEV1/FVC < 70%	1.051 (0.984–2.797)	0.055		

LC: lung cancer; CPFE: combined pulmonary fibrosis and emphysema; OR: odds ratio; CI: interval confidence; ALC: absolute lymphocyte count; CRP: C-reaction protein; IL-6: interleukin-6; C3: complement 3; C4: complement 4; FEV1/FVC%: forced expiratory volume in the first second to forced vital capacity ratio.

**Table 4 jcm-12-01100-t004:** Comparisons between the low and high C3 groups of lung cancer patients with CPFE.

	Low C3 Group (C3 < 1.00 g/L) *n* = 30	High C3 Group (C3 ≥ 1.00 g/L) *n* = 29	*p* Value
Age, years	69 (61, 78)	65 (58, 73)	0.094
Sex (men)	30/30 (100.0%)	29/29 (100.0%)	1.000
Ex- or current smokers	29/30 (96.7%)	21/29 (72.4%)	0.092
Pack-years	35 (19, 51)	30 (18, 46)	0.063
WBC, ×10^9^/L	8.44 (6.17, 10.37)	9.12 (6.01, 12.07)	0.150
ANC, ×10^9^/L	6.15 (3.85, 8.34)	6.56 (4.02, 8.94)	0.340
ALC, ×10^9^/L	1.22 (0.78–1.81)	1.62 (1.17–2.46)	0.070
PLT, ×10^9^/L	174.5 (126.5–228.5)	214.0 (114.5–261.5)	0.347
NLR	4.17 (2.87–6.99)	3.67 (2.19–5.80)	0.243
PLR	143.53 (98.46–208.94)	108.41 (191.17)	0.182
CRP, mg/L	51.90 (43.19, 60.21)	82.37 (61.89, 90.91)	0.039
IL-6, ug/L	53.69 (31.61, 57.68)	89.80 (66.57, 127.13)	0.035
Pathological type of tumor			0.441
Adenocarcinoma	13/30 (43.3%)	11/29 (37.9%)	
Squamous carcinoma	11/30 (36.7%)	7/29 (24.1%)	
Small cell lung cancer	2/30 (6.7%)	5/29 (17.2%)	
Others	5/30 (16.7%)	6/29 (20.7%)	
Location			0.749
Cancer in fibrotic areas	16/30 (58.3%)	17/29 (53.8%)	
Cancer in emphysema areas	14/30 (41.7%)	12/29 (46.2%)	
Staging (III/IV)	24/30 (80.0%)	25/29 (86.2%)	0.922
Distant metastasis	15/30 (50.0%)	17/29 (58.6%)	0.089
Contralateral lung	4/15 (33.3%)	6/17 (35.3%)	0.728
Pleura	6/15 (50.0%)	5/17 (29.4%)	0.623
Bone	4/15 (33.3%)	7/17 (41.2%)	0.382
Liver	2/15 (16.7%)	1/17 (5.9%)	0.602
Brain	0/15 (0.0%)	2/17 (11.8%)	0.059
Adrenal gland	1/15 (8.3%)	1/17 (5.9%)	0.735
Emphysema phenotype			0.084
Centrilobular emphysema	19/30 (37.5%)	15/29 (30.8%)	
Panlobular emphysema	0/30 (0.0%)	2/29 (0.0%)	
Paraspinal emphysema	11/30 (62.5%)	12/29 (69.2%)	
PA/A > 1	13/30 (43.3%)	13/29 (44.8%)	0.908

C3: complement 3; WBC: white blood count; ANC: absolute neutrophil count; ALC: absolute lymphocyte count; PLT: platelet count; NLR: neutrophil to lymphocyte ratio; PLR: platelet to lymphocyte ratio; CRP: C-reaction protein; IL-6: interleukin-6; PA/A: pulmonary artery to aorta ratio.

**Table 5 jcm-12-01100-t005:** Comparisons between CPFE patients with AE and without AE.

	AE Group *n* = 17	Without AE Group *n* = 110	*p* Value
Sex (men)	17/17 (100.0%)	101/110 (91.8%)	0.607
Age, years	65 (56, 72)	69 (64, 75)	0.057
BMI, kg/m^2^	20.88 (19.63, 24.98)	22.23 (20.03, 24.61)	0.685
Ex- or current smokers	14/17 (82.4%)	82/110 (74.5%)	0.761
Pack-years	30 (10, 45)	30 (0, 45)	0.765
Dust exposure	0/17 (0.0%)	1/110 (0.9%)	0.866
Coronary heart disease	4/17 (23.5%)	16/110 (14.5%)	0.471
Diabetes mellitus	7/17 (41.2%)	22/110 (20.0%)	0.066
Previous pulmonary tuberculosis	3/17 (17.6%)	15/110 (13.6%)	0.709
Underlying disease			0.092
Without underlying disease	8/17 (47.1%)	63/110 (57.3%)	0.184
One type of disease	4/17 (23.5%)	37/110 (33.6%)	0.081
≥2 types of underlying disease	5/17 (29.4%)	10/110 (9.1%)	0.003
ANC, ×10^9^/L	6.09 (4.21, 9.06)	5.12 (3.54, 6.58)	0.139
WBC, ×10^9^/L	7.54 (5.74, 9.38)	8.37 (6.25, 11.35)	0.116
ALC, ×10^9^/L	1.20 (0.85, 1.86)	1.25 (0.85, 1.80)	0.942
PLT, ×10^9^/L	198.0 (129.6–248.0)	168.0 (108.5–249.0)	0.296
NLR	4.66 (2.80–8.63)	4.07 (2.99–9.19)	0.290
PLR	157.4 (102.2–223.4)	103.4 (75.6–222.9)	0.030
CRP, mg/L	78.00 (13.20, 121.50)	25.10 (8.51, 80.43)	0.046
IL-6, ug/L	39.38 (10.67, 103.75)	25.64 (11.72, 58.17)	0.926
PCT, ng/mL	0.07 (0.02–0.25)	0.07 (0.03–0.15)	0.824
BNP, ng/L	304 (105, 1723)	431 (139, 1170)	0.902
Fibrinogen, g/L	4.95 (3.75, 5.31)	4.21 (3.27, 5.58)	0.679
D-dimer, mg/L FEU	2.33 (0.85–6.31)	1.16 (0.48–2.59)	0.041
C3, g/L	0.91 (0.86, 1.08)	0.93 (0.79, 1.08)	0.668
C4, g/L	0.21 (0.17, 0.24)	0.21 (0.17, 0.27)	0.440
CD4^+^ T cells, cell/uL	263 (224–680)	443 (260–568)	0.714
CD8^+^ T cells, cell/uL	216 (132–288)	284 (220–395)	0.113
CD4/CD8	1.85 (1.10–3.05)	1.33 (0.79–1.89)	0.044
CPI	41.8 (27.1, 57.8)	32.6 (21.1, 45.2)	0.427
FEV1, %pred	80.1 (62.7, 87.8)	85.4 (73.6, 92.9)	0.215
FEV1/FVC, %	79.5 (76.1, 95.5)	78.5 (69.7, 83.1)	0.519
VC, %pred	75.2 (52.5, 85.9)	85.8 (76.9, 100.8)	0.011
DLCO, %pred	39.2 (29.0, 46.0)	53.9 (45.8, 69.8)	0.000
Morphology of pulmonary fibrosis			0.178
Typical honeycombing	9/17 (52.9%)	52/110 (47.3%)	
Reticular pattern	5/17 (29.4%)	46/110 (41.8%)	
Reticular pattern with mild GGO	3/17 (17.6%)	12/110 (10.9%)	
Peripheral traction bronchiectasis or bronchiolectasis	14/17 (82.4%)	64/110 (58.2%)	0.057
Emphysema phenotype			0.082
Paraseptal emphysema	13/17 (76.5%)	56/110 (50.9%)	0.058
Panlobular emphysema	1/17 (5.9%)	5/110 (4.5%)	0.877
Centrilobular emphysema	3/17 (17.6%)	49/110 (44.5%)	0.021
PA/A > 1	10/17 (58.8%)	55/110 (50.0%)	0.604
MPA, mm	26.5 (21.3–27.3)	23.0 (21.5–27.0)	0.596
Lung cancer diagnosis	6/17 (35.3%)	53/110 (48.2%)	0.183

BMI: body mass index; underlying disease: including diabetes mellitus, hypertension, coronary heart disease, pulmonary tuberculosis, dust exposure, end-stage renal disease, liver cirrhosis, autoimmune disorders and hematological disease; ANC: absolute neutrophil count; WBC: white blood count; ALC: absolute lymphocyte count; PLT: platelet count; NLR: neutrophil to lymphocyte ratio; PLR: platelet to lymphocyte ratio; CRP: C-reaction protein; IL-6: interleukin-6; PCT: procalcitonin; BNP: brain natriuretic peptide; FEU: fibrinogen equivalent units; C3: complement 3; C4: complement 4; CPI: compound physiological index; FEV1: forced expiratory volume in the first second; FVC: forced vital capacity; FEV1/FVC%: forced expiratory volume in the first second to forced vital capacity ratio; VC: vital capacity; DLCO: diffusing capacity for carbon monoxide; GGO: ground-glass opacity; PA/A: pulmonary artery to aorta ratio; MPA: main pulmonary artery; AE: acute exacerbation.

**Table 6 jcm-12-01100-t006:** Risk factors for AE in patients with CPFE.

Variable	Univariate Analyses		Multivariate Analyses	
	HR (95% CI)	*p* Value	HR (95% CI)	*p* Value
Age, years	1.040 (0.918–1.178)	0.535		
Diabetes mellitus	2.800 (0.958–8.187)	0.060		
Underlying disease: without	Re			
One type of disease	0.851 (0.240–3.023)	0.803		
≥2 types of underlying disease	3.580 (0.987–12.976)	0.052		
PLR	1.619 (1.011–2.593)	0.045	3.731 (1.288–10.813)	0.015
CRP, mg/L	1.004 (0.999–1.023)	0.083		
D-dimer, mg/L FEU	1.034 (0.947–1.128)	0.459		
CD4/CD8	1.001 (0.998–1.004)	0.426		
VC, %pred	0.283 (0.011–0.617)	0.031	0.577 (0.137–0.918)	0.027
DLCO, %pred	0.783 (0.221–0.861)	0.057	0.919 (0.863–0.979)	0.009
Peripheral traction bronchiectasis or bronchiolectasis	3.354 (0.911–12.347)	0.069		
Emphysema phenotype: other type	Re			
Paraseptal emphysema	3.714 (0.999–13.811)	0.050		
Lung cancer diagnosis	3.981 (0.435–5.440)	0.876		

HR: hazards regression; CI: interval confidence; Underlying disease: including diabetes mellitus, hypertension, coronary heart disease, pulmonary tuberculosis, dust exposure, end-stage renal disease, liver cirrhosis, autoimmune disorders and hematological disease; PLR: platelet to lymphocyte ratio; CRP: C-reaction protein; VC: vital capacity; DLCO: diffusing capacity for carbon monoxide; AE: acute exacerbation; Re: reference.

## Data Availability

The datasets used and/or analyzed during the current study are available from the corresponding author on reasonable request.

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
