# Peer review of "The Impact of Lung Cancer in Patients with Combined Pulmonary Fibrosis and Emphysema (CPFE)"

_jcm, 2023, doi:10.3390/jcm12031100_

Round 1

Reviewer 1 Report (Previous Reviewer 1)

Congratulations 

Author Response

It is a great honor for us to accept comments and suggestions from reviewers on the manuscript. We deeply appreciate your consideration and recognition of our manuscript, It is hoped that our study will contribute to better clinical management of CPFE patients with LC, thereby avoiding missed and delayed diagnosis and reducing mortality. We sincerely hope that the correction will meet with approval. If there are any errors or other advice on this manuscript, please do not hesitate to tell us your valuable comments and suggestions, which will be a generous help to offer further supplementary and perfection for our study. Thank you very much. We look forward to receiving comments from the reviewers. 

Reviewer 2 Report (Previous Reviewer 2)

In this paper Feng and colleagues described a population of patients presenting both Combined Pulmonary Fibrosis and Emphysema (CPFE) and Lung Cancer (LC). This population was described and compared to a population of CPFE patients without LC. Multiple values were screened to understand their association to LC and C3 complement blood levels was found as a possible marker. The population was then sub grouped in high and low C3 accordingly to a Youden test. The high C3 population was compared both in the overall population than only in the LC population. The population was then followed up for at least 2 years for Acute Exacerbations (AE) to understand the role of LC in the development of AE.

I appreciate the work of the authors to address a topic that is underrated in the literature and especially I appreciate how well they have reviewed the paper; however, re-reading this paper lead to some other points that I would like to rise to the authors to furthermore improve their work.

Major suggestions

-        Relation between simple values as lymphocytes, monocytes, platelets or their ratios with the presence or the prognosis of LC in interstitial lung diseases has been already studied (https://doi.org/10.1016/j.rmed.2021.106686) ad as prognostic factors for ILDs (https://doi.org/10.3390/medicina56080381.; https://doi.org/10.1007/s00408-020-00386-7.; doi:10.1164/rccm.202003-0669oc.; https://doi.org/10.1177/1479973120909840.) . Authors should try if any of this values are associated to lung cancer or prognostic for AE in their population.

-        The current definition of Acute Exacerbation of IPF is different from the one used by the authors (see Collard et al. doi: 10.1164/rccm.201604-0801CI.): not only idiopathic exacerbations are to take in consideration, but also the ones triggered by infections or other causes. I agree with the authors to exclude chemotherapy and surgical triggered AE, but infective or other causes should be included.

-        As already stated, odd ratio cannot be used as predictors, OR are values of association, especially in a population were LC and CPFE are diagnosed in contemporaneity. All the references to prediction/incidence/developmentein the correlation between clinical data and LC should be amended with association/prevalence/presence, as well written in the conclusions.

-        Table 4 should be implemented with more values that are possible risk factors for AE at CPFE diagnosis.

Minor suggestions

-        In the methods section is not clear that the authors only selected patients with a UIP definite/probable pattern in the lower lobes. I suggest to clarify that the definition of CPFE includes all of fILD patterns (as defined in ‘Syndrome of Combined Pulmonary Fibrosis and Emphysema An Official ATS/ERS/JRS/ALAT Research Statement’ by Cottin et al. published in May 2022) but the authors excluded patterns different from UIP.

-        I do not understand why in Table 3 many OR values not overlapping the 0 are not statistically significant. Authors should revaluate this table.

-        Authors describe as obvious that CPFE patients with reduced DLco values are more prone to AE, however, I believe this isn’t so obvious, even if is known in literature that worst functional values are related to major incidence of AE in IPF.

Round 2

Reviewer 2 Report (Previous Reviewer 2)

I thank the authors for the great work made on this paper

This manuscript is a resubmission of an earlier submission. The following is a list of the peer review reports and author responses from that submission.

Round 1

Reviewer 1 Report

This is an interesting study, although many results are difficult to apply in clinical settings. Still, this study has been done with hard work and is written well. It is good that the binary logistic method has been used. The ROC curve has been plotted to find the cutoff for C3.

In Fig 2 there is something written under non-smokers written in some other language which should be corrected

Reviewer 2 Report

In this paper Feng and colleagues described a population of patients presenting both Combined Pulmonary Fibrosis and Emphysema (CPFE) and Lung Cancer (LC). This population was described and compared to a population of CPFE patients without LC. Multiple values were screened to understand their role as risk factors for LC and C3 complement blood levels was found as a possible marker. The population was then sub grouped in high and low C3 accordingly to a Youden test. The high C3 population was compared both in the overall population than only in the LC population. The LC was also grouped by the histology of the LC to compare squamous cell LC to adenocarcinoma.

I appreciate the work of the authors to address a topic that is underrated in the literature; however, the study presents some methodological biases and some errors in the conclusions that are only partially supported by the data.

I would like the authors to work on the following points:

-     - Authors state in the methods section (section 2.2) that the CPFE criteria used are the one proposed by Cottin in his paper from 2005 (cit 1): presence of emphysema and UIP pattern. However, firstly, Cottin does not provide a definition of CPFE in that paper, secondly the current definition of CPFE in literature is “presence of classic features of centrilobular and/or paraseptal emphysema in the upper lobes and pulmonary fibrosis in the lower lobes” as stated in cit. 3. This different definition in the paper and the multiple citation trough the paper to “CPFE patients with UIP” creates confusion. I suggest the authors to revise the population in a multidisciplinary team composed at least of a pulmonologist and a radiologist expert in thoracic radiology to give a definition of the lower lobes ILD. This data should be added to Table 2 with a comparison between the two populations.

-   Connected to the above point CPFE diagnosis should not be based only by two expert clinicians but must be made in a multidisciplinary discussion.

-    Multiple values (ANA title, C3 and probably others) have a high percentage of missing data. Those values presenting more than 20% of missing data should not be added to the logistic regression. For this reason all the evaluation about low and High C3 must be cancelled from the paper, or, at least, the authors should rewrite the paper presenting the data only about the patients with this value, redoing all the statistical analysis.

-  Authors must eliminate in the abstract and the conclusions the mention to non-statistically significant data, such as an higher probability of metastasis (especially brain) or the correlation between squamous cell LC to emphysema areas.

Some minor points

-  In the methods section is not clear if the control population (CPFE without LC) were newly diagnosed, the authors should clarify this in the methods section.

-  In table 1 is not clear why 124 patients are listed for the staging when only 122 patients are enrolled, and on this topic I would like to have a Table presenting how the patients were staged (CT scans, PET-CT, mediastinal EBUS)

- Only 50 patients present a degree of differentiation, due to the lack of data this result should be removed both from table 1 and the text.

- It is not clear how the authors count Pack-Years. The current definition is “It is calculated by multiplying the number of packs of cigarettes smoked per day by the number of years the person has smoked. For example, 1 pack year is equal to smoking 1 pack per day for 1 year, or 2 packs per day for half a year, and so on.” In this study patients smoked a mean of 733 P/Y and had 68 Years, and the count is not realistic and should be amended.

- Authors could not use a odd ratio as a risk factor, lacking of the time period (e.g. we cannot know if the high C3 value is a risk factor for the LC or is due to the presence of the LC, we only know that they are independently linked).

- The paper should be checked with the STROBE checklist to be sure to be aligned to it